# Orientation of Chiral Schiff Base Metal Complexes Involving Azo-Groups for Induced CD on Gold Nanoparticles by Polarized UV Light Irradiation

**Nobumitsu Sunaga, Tomoyuki Haraguchi and Takashiro Akitsu \***

Department of Chemistry, Faculty of Science, Tokyo University of Science, 1–3 Kagurazaka, Shinjuku-ku, Tokyo 162-8601, Japan
\* Correspondence: akitsu2@rs.tus.ac.jp; Tel.: +81-3-5228-8271

**Abstract:** In this study, we report the synthesis, characterization, and chiroptical properties of azo-group-containing chiral salen type Schiff base Ni(II), Cu(II), and Zn(II) complexes absorbed on gold nanoparticles (AuNPs) of 10 nm diameters. Induced circular dichroism (CD) around the plasmon region from the chiral species weakly adsorbed on the surface of AuNP were observed when there were appropriate dipole–dipole interactions at the initial states. Spectral changes were also observed by not only *cis-trans* photoisomerization of azo-groups but also changes of orientation due to Weigert effect of azo-dyes after linearly polarized UV light irradiation. Spatial features were discussed based on dipole-dipole interactions mainly within an exciton framework.

**Keywords:** chirality; azobenzene; photo-isomerization; Weigert effect; Schiff base complexes; gold nanoparticles; time-dependent density functional theory (TD-DFT)

## 1. Introduction

Despite having an identical chemical formula, some compounds show differences in chemical properties or optical properties. Such molecules are called "chiral" molecules, which may be one of the most important concepts concerning biomolecules and stereochemistry. There is no difference in physical properties between a pair of enantiomer molecules in a mirror image relationship. In this context, differences in the magnitude of light absorption are conventionally called circular dichroism (CD) [1].

Recently, beyond chiral metal nanoparticles [2,3], chiral hybrid materials composed of chiral molecules and (achiral) semiconductor or metal nanoparticles, such as gold nanoparticles (AuNP), have been studied [3–5] and developed. The hybrid systems of AuNP and chiral metal complexes have been investigated for biomaterials [6–8]. When chiral metal compounds are adsorbed on AuNP (or nanostructured surface of gold substrates), interaction of the chiroptical signals between their dipole moments may be caused by surface plasmon resonance [4,5,9]. When they have (chemically) specific bonds, the magnitude of induced CD spectra increases or decreases depending on the direction of the dipole moments. Some types of exciton mechanisms have been proposed for theoretical predictions or experimental explanation [10,11]. The importance of both magnitude and direction (phenomenologically speaking, parallel arrangement) of electric transition dipole moments (longer molecular axis) of some chiral metal complexes and (vertical vector of) AuNP has also been investigated experimentally [12–14].

We have investigated photofunctional hybrid materials of some (chiral/achiral) Schiff base metal complexes and photochromic azobenzene, and their interaction with several types of topological lights such as polarized UV light [15–20]. In addition, we have studied the theoretical interpretation of the

optical properties of these materials [21–25]. In particular, we focused on the molecular orientation due to the dipole-dipole interaction between two molecules. For example, there are some studies on the molecular orientation of photofunctional organic molecules in polymer films [2].

In this study, azo-group-containing chiral salen-type Ni(II), Cu(II), and Zn(II) complexes were synthesized to control the molecular interaction by small differences. To prove the roles of dipole moment and environment, steric factors of intermolecular interaction were designed for the ligands [26]. In solutions of various complexes and colloidal AuNPs (or nanomaterials) [27–29], the influence of photoisomerization due to azo-group on the changes of induced CD bands is investigated by comparison to previous data. New types of molecular re-orientation could be proposed according to the interpretation of induced CD bands in the plasmon wavelength region of AuNP. Furthermore, we herein demonstrate that linearly polarized UV light (Weigert effect) [30,31] inducing novel motion of the azo-compounds on the surface of AuNPs.

## 2. Materials and Methods

### 2.1. General Procedures and Preparations

Chemicals and solvents of the highest commercial grade available from Kanto Chemical (Tokyo, Japan) for solvents, from Tokyo Chemical Industry (Tokyo, Japan) for organic compounds, from Wako (Osaka, Japan) for metal sources, and Funakoshi (Tokyo, Japan) for AuNP (10 nm) were used as received without further purification. Cu(II), Ni(II), and Zn(II) complexes (CuAz, NiAz, and ZnAz) were prepared according to published methods using corresponding *n*-propyl compounds instead of ethyl compounds (Figure 1) [14]. Mixed methanol solutions (1:1, *v/v*) for tests were prepared using 0.002 mM complexes and commercially available AuNP solutions (purchased from Funakoshi).

**Figure 1.** Preparation scheme of CuAz, NiAz, and ZnAz.

### 2.2. Characterization of Complexes

The complexes obtained were conventionally characterized as follows. In the order of the number of identified solvent molecules in a crystal for each complex, elemental analysis was carried out as the corresponding hydrates.

NiAz: yield 82%. Anal. Found: C, 64.29; H, 4.84; N, 9.28%. Calc. for $C_{48}H_{50}N_6NiO_{10}$ (as tetrahydrate) C, 64.52; H, 5.19; N, 9.40%. IR (KBr (cm$^{-1}$)): 1464, 1535 (N=N), 1605 (C=N).

CuAz: yield 83%. Anal. Found: C, 66.16; H, 4.93; N, 9.60%. Calc. for $C_{48}CuH_{44}N_6O_7$ (as monohydrate) C, 66.34; H, 4.79; N, 9.67%. IR (KBr (cm$^{-1}$)): 1461, 1530 (N=N), 1625 (C=N).

ZnAz: yield 79%. Anal. Found: C, 64.87; H, 5.14; N, 9.37%. Calc. for $C_{48}H_{48}N_6O_9Zn$ (as trihydrate) C, 64.93; H, 5.11; N, 9.46%. IR (KBr (cm$^{-1}$)): 1468, 1533 (N=N), 1630 (C=N).

*2.3. Physical Measurements*

Elemental analysis was carried out with a Perkin-Elmer 2400II CHNS/O analyzer (Perkin-Elmer, Waltham, USA) at Tokyo University of Science. Infrared (IR) spectra were recorded on a JASCO (Tokyo, Japan) FT-IR 4200 spectrophotometer (JASCO, Tokyo, Japan) in the range of 4000–400 cm$^{-1}$ at 298 K. Electronic (UV-vis) spectra were measured on a JASCO V-650 spectrophotometer (JASCO, Tokyo, Japan) equipped with polarizer in the range of 800–220 nm at 298 K. Circular dichroism (CD) spectra were measured on a JASCO J-725 spectropolarimeter (JASCO, Tokyo, Japan) in the range of 800–200 nm at 298 K. Photo-illumination experiments were carried out using a lamp (1.0 mW/cm$^2$) by Hayashi Tokei co. ltd. (Tokyo, Japan) with optical filters (UV λ = 200–400 nm) leading to a sample using optical fibers and a polarizer through optical filters.

*2.4. Computational Methods*

Calculations for all complexes were performed using the Gaussian 09W software (Revision D.02, Gaussian, Inc., Wallingford, CT, USA) [32]. All the geometries were optimized using B3LYP level of theory and SDD as basis set. Furthermore, we performed frequency calculations on the optimized geometry using the same level of theory and basis set.

## 3. Results and Discussion

*3.1. Simulated CD Spectra with TD-DFT*

First, we computationally confirmed the molecular-level chirality of the complexes. Optimized structures for CuAz, NiAz, and ZnAz (*trans*-forms about azo-groups) were obtained by means of TD-DFT calculations. Simulated CD and UV-vis spectra (Figure 2) and calculated numerical data (Tables 1 and 2) were obtained from the optimized structures of *trans*-forms and utilized to assign spectral peaks in the next section. Overall electric dipole moment would be along the metal-to-middle point of chiral centers direction similar to the analogous chiral salen-type complexes [12]. Compared to a previous study [14], optimized structures offer a (slightly distorted) square planar coordination geometry with slightly steric differences due to the ligand's terminal *n*-propyl-groups. Overall spectral features and assignment of predominant peaks were also similar to other azo-containing complexes [33]. Expected spectral changes due to *trans*- to *cis*-photoisomerization by UV light irradiation appeared below ~400 nm. Only the ZnAz complex is diamagnetic, does not show d-d transition, and may afford a slightly distorted (compressed) tetrahedral coordination geometry. Therefore, agreement of calculated spectra, spectral feature of visible region, and change of induced spectra (caused by adsorption form) may differ from others. Due to ligand symmetry, the dipole moment's direction will be kept but its magnitude will change after azo-groups' photoisomerization [34]. However, if photo-isomerization is expected, the electric dipole moment of the complex will decrease. Conventionally, the optical activity (optical rotation intensity) of chiral molecules is proportional to the imaginary part of the inner product of the intramolecular electric (μ) and magnetic (m) transition dipole moments. Electrical and magnetic transition dipole moments were calculated using TD-DFT theoretical calculations to determine the optimized structure of each complex as the *trans*-form (Tables 1 and 2). In order to discuss only *trans*-forms (differences as *cis*-forms and molecular orientation will be discussed later), only the calculation results of the *trans*-form were compared to the experimental spectrum to confirm the validity of structure optimization.

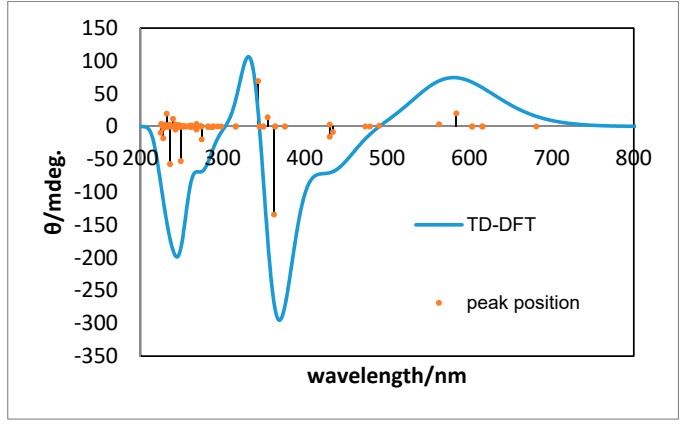

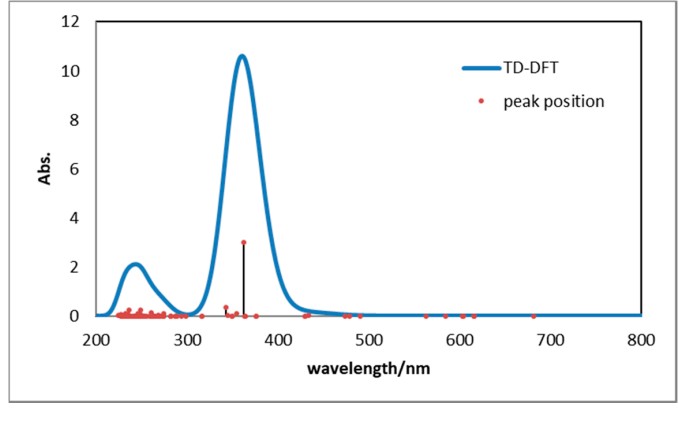

(**a**)

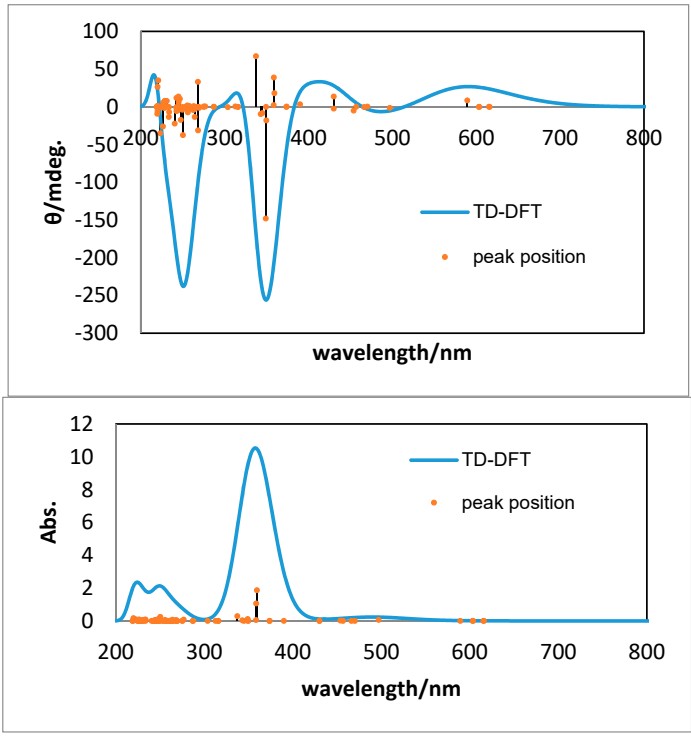

(**b**)

**Figure 2.** *Cont.*

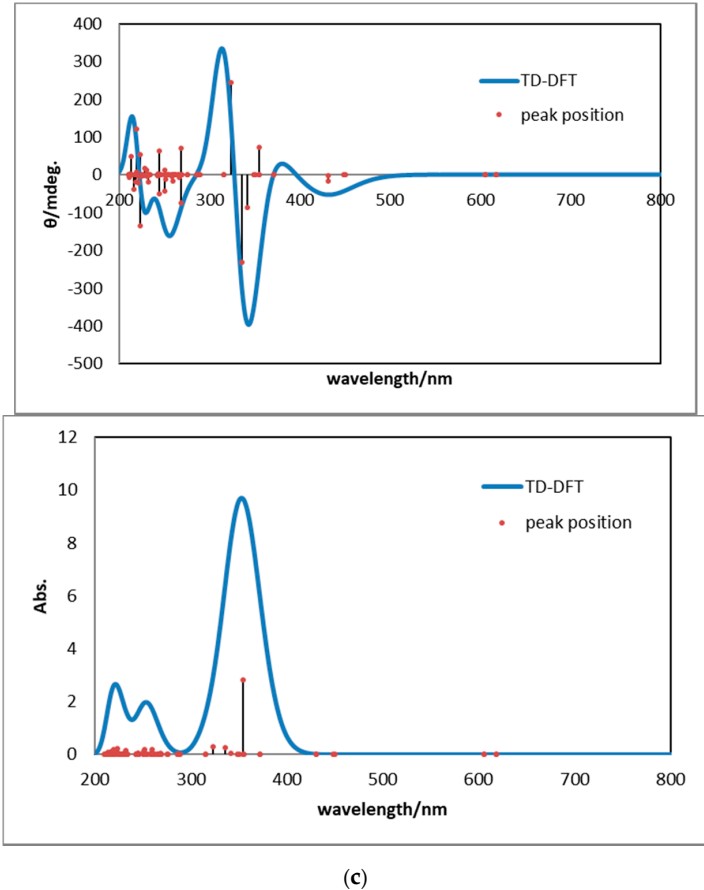

(**c**)

**Figure 2.** Simulated CD and UV-vis spectra for (**a**) NiAz, (**b**) CuAz, and (**c**) ZnAz.

**Table 1.** Calculated data about transition movements of visible region peaks for *trans*-forms of NiAz, CuAz, and ZnAz.

| Wavelength/nm | Oscillator Strength | Dipole Strength | Electric Transition Dipole Moment μ (x,y,z) | | | Magnetic Transition Dipole Moment m (x,y,z) | | |
|---|---|---|---|---|---|---|---|---|
| NiAz | | | | | | | | |
| 584.12 | 0.0006 | 0.0107 | 0.1036 | 0.0001 | −0.0018 | −0.8206 | 0.0002 | 0.0571 |
| 563.16 | 0.0009 | 0.0162 | 0.1273 | 0.0002 | 0.0051 | −0.1091 | −0.0035 | 0.0142 |
| 433.97 | 0.043 | 0.6148 | −0.7839 | 0.0003 | 0.016 | −0.0761 | −0.0245 | −1.5352 |
| 430.13 | 0 | 0.0002 | −0.0135 | −0.0026 | −0.0017 | 0.8302 | 0.4876 | 0.0093 |
| CuAz | | | | | | | | |
| 589.19 | 0.0008 | 0.0154 | 0.1241 | −0.0001 | −0.0018 | −0.2885 | −0.0046 | 0.1496 |
| 496.86 | 0.0635 | 1.0390 | −1.0193 | 0.003 | 0 | −0.0057 | −0.0153 | −0.4703 |
| 470.13 | 0.0002 | 0.0034 | 0.0584 | −0.0013 | −0.0011 | −0.0184 | −0.0068 | 0.0367 |
| 456.79 | 0.0001 | 0.0012 | 0.0352 | −0.0014 | −0.0009 | 0.0051 | 0.55 | 0.0917 |
| 453.76 | 0.0207 | 0.3096 | 0.5563 | −0.0005 | −0.0122 | 0.053 | −0.0644 | 0.6525 |
| ZnAz | | | | | | | | |
| 431.06 | 0.0002 | 0.0030 | −0.0544 | 0 | −0.0033 | −1.2446 | 0.0042 | 0.0887 |
| 431.06 | 0 | 0.0000 | 0.0004 | 0.0058 | 0 | 0.0093 | 0.5579 | −0.0007 |

**Table 2.** Calculated data regarding the optical rotation intensity of predominant peaks for *trans*-forms of NiAz, CuAz, and ZnAz.

| Wavelength/nm | Optical Rotation Intensity | $\mu \cdot m$ | $\lvert \mu \rvert$ | $\lvert m \rvert$ | $\lvert \mu \rvert \, \lvert m \rvert$ | $\cos\theta$ |
|---|---|---|---|---|---|---|
| NiAz | | | | | | |
| 430.08 | −0.03283938 | 0.06637402 | 0.07181685 | 0.97539311 | 0.070049661 | −0.468801412 |
| 354.63 | 0.0293979 | −0.05934276 | 1.060564534 | 2.911695934 | 3.08804144 | 0.009519918 |
| CuAz | | | | | | |
| 389.85 | 0.00701379 | −0.01416759 | 0.062957367 | 0.281213673 | 0.017704472 | 0.396159224 |
| 359.29 | 0.03800265 | −0.07681704 | 4.701237033 | 0.653232891 | 3.071002659 | 0.012374672 |
| ZnAz | | | | | | |
| 431.06 | −0.0016 | 0.00324 | 0.005814 | 0.557977948 | 0.003244 | −0.49445 |
| 354.56 | 0.15520407 | −0.31356246 | 5.718414271 | 1.067998895 | 6.107260124 | 0.025413044 |
| 323.4 | 0.51463755 | −1.03966038 | 1.7919 | 0.5802 | 1.03966038 | 0.495005446 |

Due to the difficulty of treating excited states by any methodsof theoretical calculations such as DFT and semi-empirical methods of normal accuracy, simulated CD and UV-vis spectra may be reliable in a qualitative sense. Therefore, it should be noted that there is a limit to the quantitative discussion as well as the detailed comparison between experimental and simulated CD and UV-vis spectra.

Generally, spectroscopic study in solutions is to treat compounds as they are in solutions, which is not necessarily identical to forms as isolated in crystalline solid-state. The reasons may be including crystalline solvents or as axial ligands in a crystal, which can be easily released when dissolved solutions. Moreover, molecular conformation of a certain metal complex, such as the present ones having long pendant ligands in a crystal, can flexibly change into other conformation in a solution. From a viewpoint of heat of formation and intermolecular interactions, molecular conformation in a solution is more stable than that in a crystal. Experimentally, however, spectra in solutions and molecular structures distorted by crystal packing can be only obtained. Therefore, computational simulation play an important role in discussing the (assumed) molecular forms in solutions to compare with or discuss spectra in solutions.

*3.2. CD and UV-vis Spectra before Irradiation*

Figures 3–5 exhibit experimental CD and UV-vis spectra before UV light irradiation for NiAz, CuAz, and ZnAz (and AuNP), respectively. For all UV-vis spectra, the surface plasmon peaks of AuNP were observed at 520 nm. As for NiAz (Figure 3), CD peaks appeared at 280 nm ($\pi$-$\pi$*) and 410 nm (n-$\pi$*), while UV-vis peaks appeared at 260 nm ($\pi$-$\pi$*) and 380 nm (n-$\pi$*). Broad and negative induced CD peaks were observed around 525 nm corresponding to the plasmon region. As for CuAz (Figure 4), CD peaks appeared at 300 nm ($\pi$-$\pi$*) and 410 nm (n-$\pi$*), while UV-vis peaks appeared at 250 nm ($\pi$-$\pi$*) and 370 nm (n-$\pi$*). A positive-induced CD peak was observed around 530 nm corresponding to the plasmon region. As for ZnAz (Figure 5), CD peaks appeared at 360 nm ($\pi$-$\pi$*) and 430 nm (n-$\pi$*), while UV-vis peaks appeared at 255 nm ($\pi$-$\pi$*) and 380 nm (n-$\pi$*). A negative-induced CD peak was observed around 550 nm corresponding to the plasmon region. According to previous studies [13,14], the intensity of d-d (weak) and (induced) plasmon bands differ from each other.

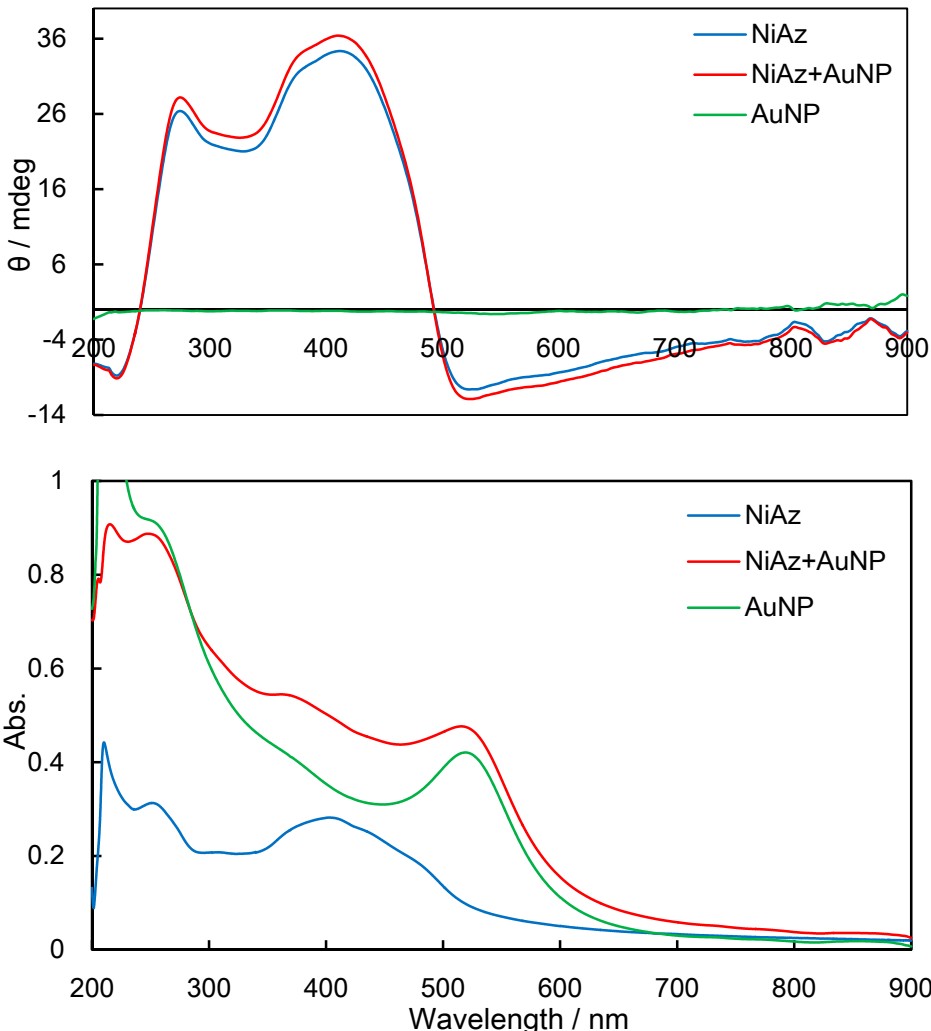

**Figure 3.** CD and UV-vis spectra for NiAz before irradiation.

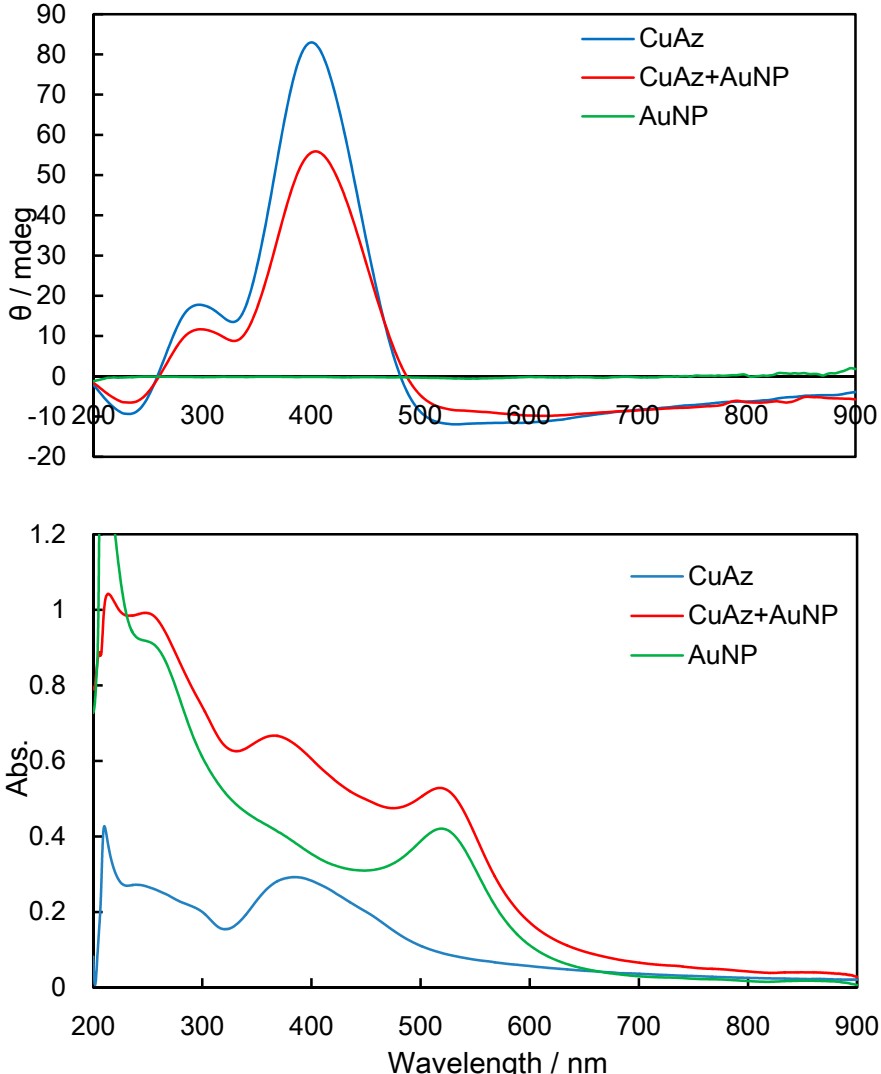

**Figure 4.** CD and UV-vis spectra for CuAz before irradiation.

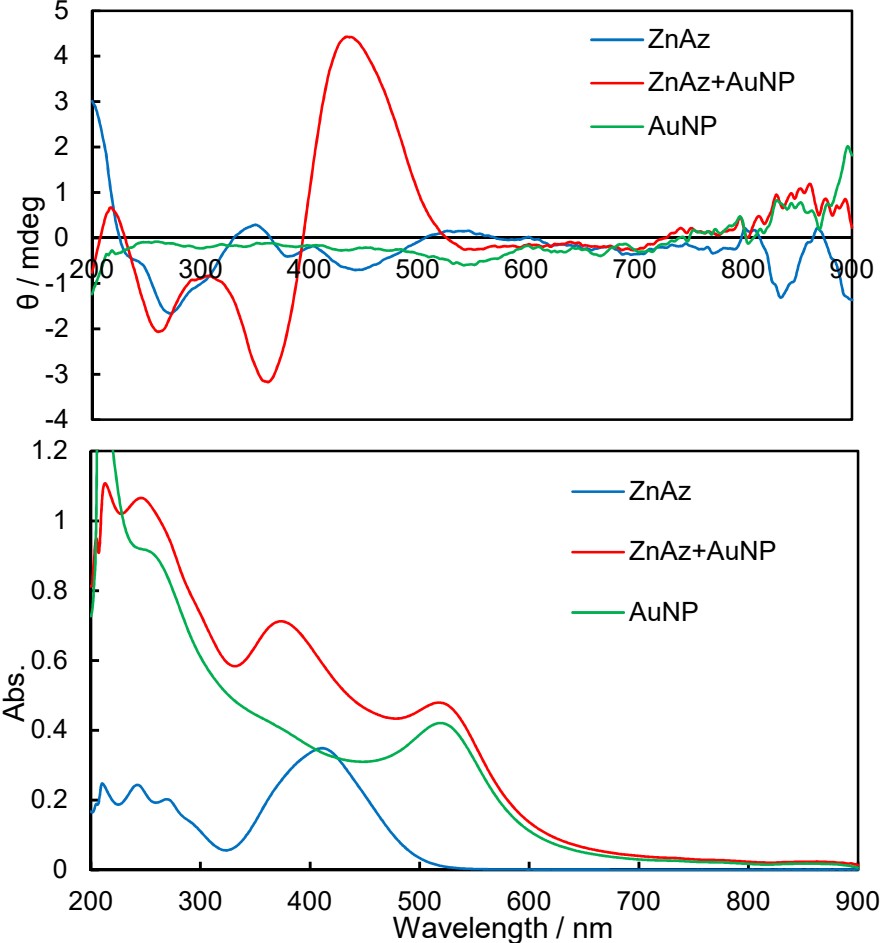

**Figure 5.** CD and UV-vis spectra for ZnAz before irradiation.

Regardless of the slight changes in ligand terminal substituent groups, the induced CD was observed very weakly in the plasmon region (around 550 nm) of gold nanoparticles. Little differences could be observed among three metals not for d-d bands but for the induced CD bands. Lack of chemical groups adsorbing onto the gold surface leads to intermolecular electromagnetic interaction between metal complexes and AuNPs, which may deny CD bands due to the direct bonding of chiral ligands. The weak adsorption structure is predicted (Figure 6) based on the theory of angles between dipole moments, originally evaluated using inner product of vectors ($\mu \cdot m$) [4]. As depicted in Figure 6, that the complexes' molecular planes were not completely parallel to the normal vector of the gold surface [13,14]. If bulkier substituents prevent chemically-bonded adsorption to induce CD bands, which may be phenomenologically improper adsorption [12,35] at least within theoretical framework using $\mu \cdot m$. For example, a prominent CD peak is observed, which may be attributed to a chiral complex itself. Adsorption structures were proposed according to CD and previous studies [13,14]. Moreover, coordination geometries, namely coordination of solvent molecules, is not clear, though the possibility of axial coordination in the solution may be high according to conventionally known solution paramagnetism of Schiff base (in particular Ni(II)) complexes. However, the presence of axial ligands may not influence the experimental results of induced CD bands.

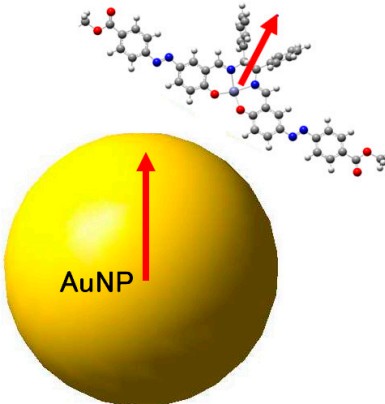

**Figure 6.** Proposed molecular arrangement of AuNP and a metal complex with their dipole moments (arrows).

### 3.3. CD and UV-vis Spectra after Irradiation

Figures 7–9 show CD and UV-vis spectra after linearly polarized UV light (<350 nm) irradiation for 5 min for NiAz, CuAz, and ZnAz (and AuNP), respectively. According to a previous study [13,14], photoismerization was generally completed at least within 3 min, which was confirmed several times. For all UV-vis spectra, surface plasmon peaks of AuNP were also observed at 520 nm. As for NiAz (Figure 7), CD peaks appeared at 280 nm (π-π* of azo group) and 410 nm (n-π* of azo group), while UV-vis peaks appeared at 260 nm (π-π* of azo group) and 360 nm (n-π* of azo group). A broad and negative induced CD peak was observed around 520 nm corresponding to the plasmon region. As for CuAz (Figure 8), CD peaks appeared at 300 nm (π-π* of azo group) and 410 nm (n-π* of azo group), while UV-vis peaks appeared at 255 nm (π-π* of azo group) and 365 nm (n-π* of azo group). A broad and positive induced CD peak was observed around 540 nm corresponding to the plasmon region. As for ZnAz (Figure 9), CD peaks appeared at 370 nm (π-π* of azo group) and 440 nm (n-π* of azo group), while UV-vis peaks appeared at 245 nm (π-π* of azo group) and 375 nm (n-π* of azo group). A negative induced CD peak was also observed around 550 nm corresponding to the plasmon region. Inter-ligand transitions (n-π* and π-π*) associated with azobenzene are quite strong bands, thus change in induced CD bands may be weak with different wavelengths. In addition, if photoisomerization can happen smoothly, induced CD bands strongly depend on adsorption structures of chiral complexes. Which are in agreement with the previous studies [13,14]. As suspension-like situations rather than clear solutions, UV-vis spectra are less sensitive than CD spectra (intense induced plasmon region) for detecting conformational changes of chiral metal complexes.

In order to clarify these differences before and after UV irradiation, Figure 10 summarizes the different spectra for the three complexes. Only for CuAz exhibited negative change (decreasing) of intensity for CD peaks, though spectral shapes (induced CD bands) were not similar to each other regardless of the identical ligand. Which may be attributed to the initial adsorption forms as well as slight differences in the coordination environments of metal complexes. As depicted in Figure 11 schematically, *trans-* to *cis*-photoisomerization of salen-type complexes resulted in different steric features of chiral complexes and their adsorption forms, as well as different electrostatic properties (decreasing of dipole moment). In contrast, *trans*-type Schiff base complexes may keep them or at least exhibit small differences because of their symmetry. Regardless, no complexes indicated CD bands before and after UV light irradiation for the present systems.

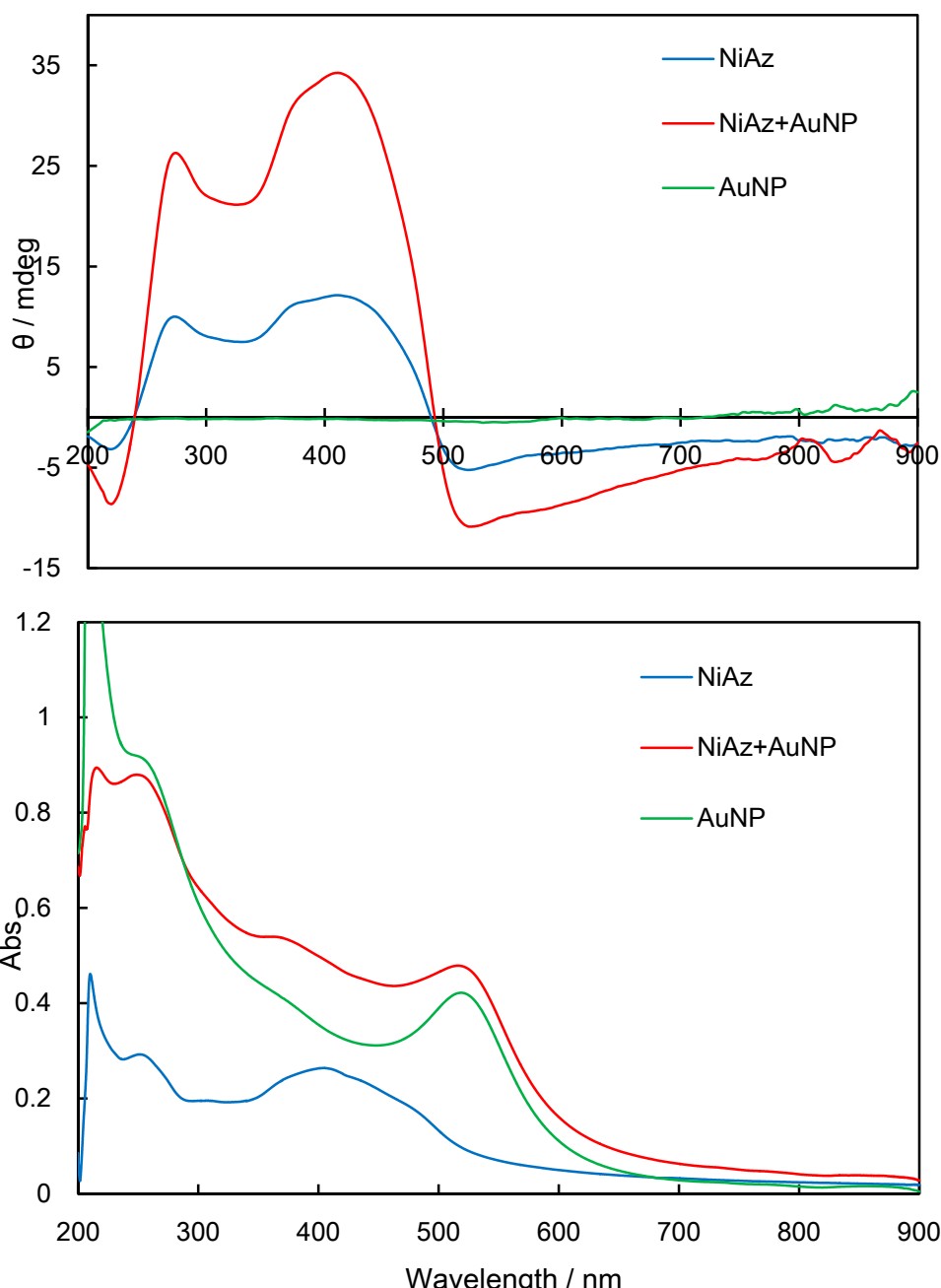

**Figure 7.** CD and UV-vis spectra for NiAz after irradiation.

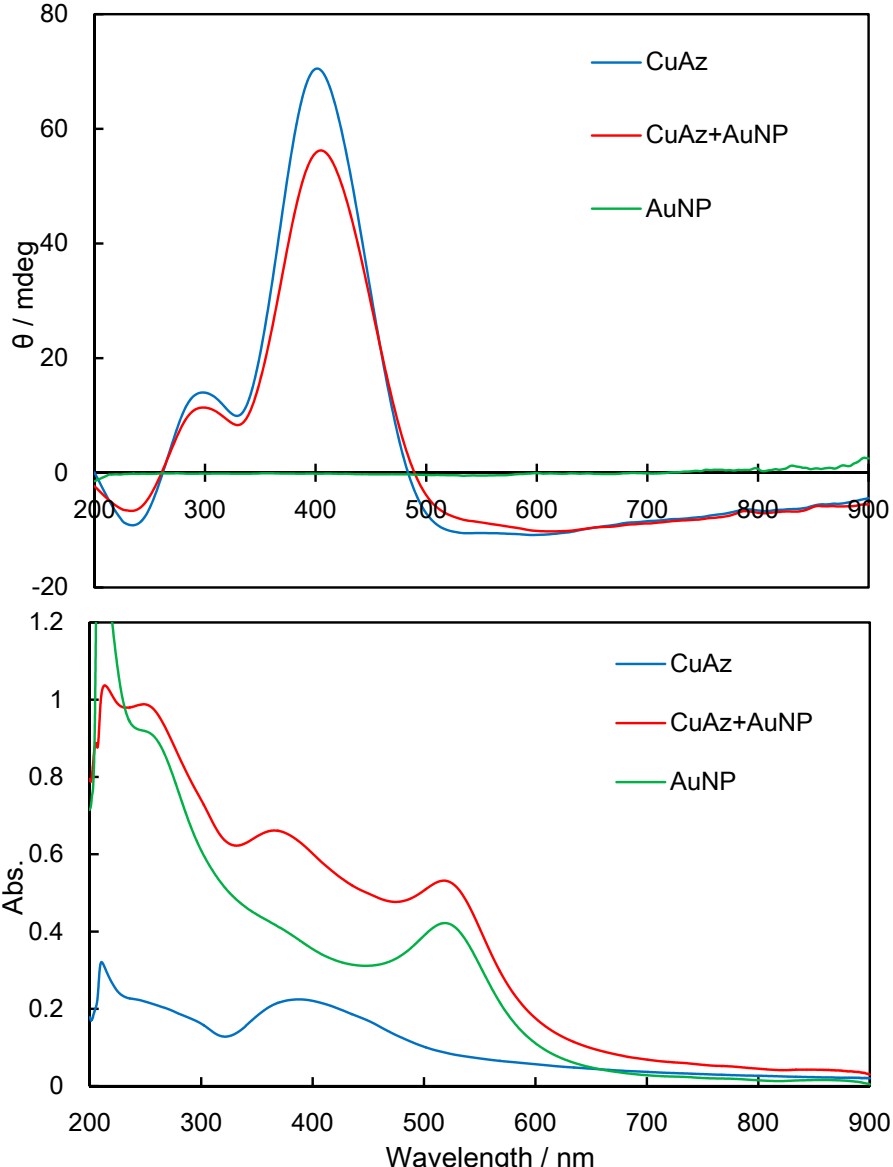

**Figure 8.** CD and UV-vis spectra for CuAz after irradiation.

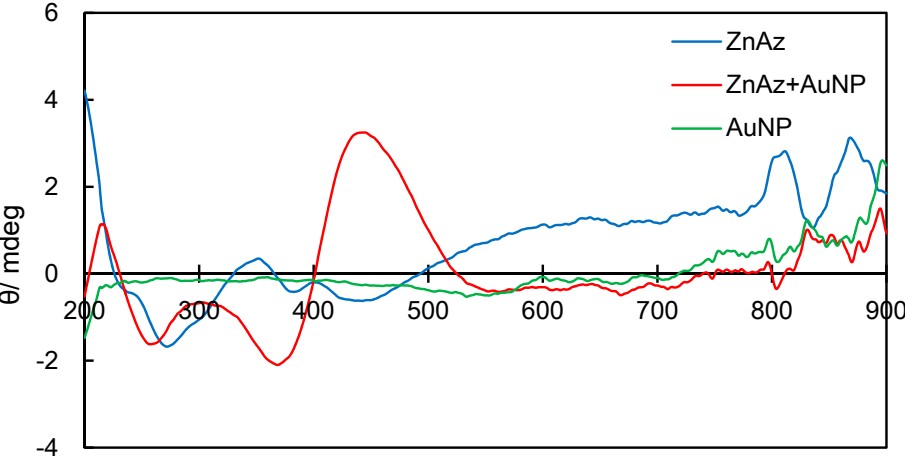

**Figure 9.** *Cont.*

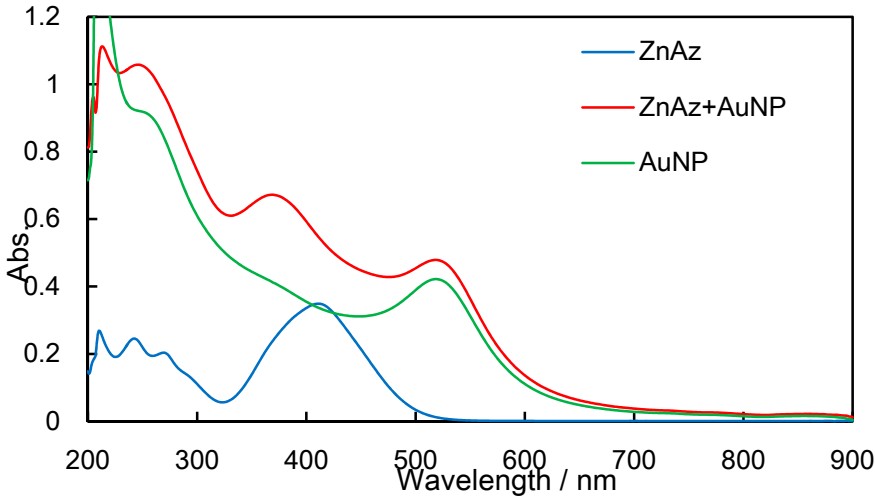

**Figure 9.** CD and UV-vis spectra for ZnAz after irradiation.

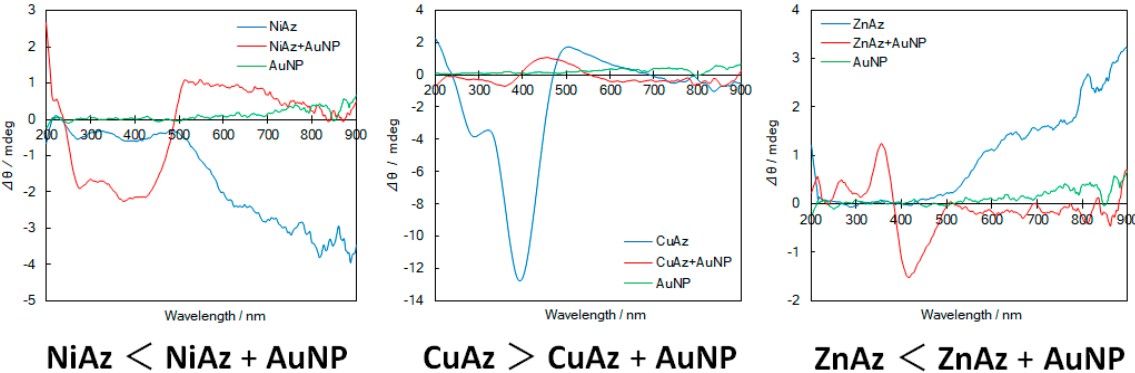

**Figure 10.** Difference in the intensity CD spectra and their sign (magnitude) for three systems before and after UV light irradiation. The sign of inequality denotes comparison of equation values $\Delta\theta$ = (CD after linearly polarized UV irradiation) - (CD before linearly polarized UV irradiation).

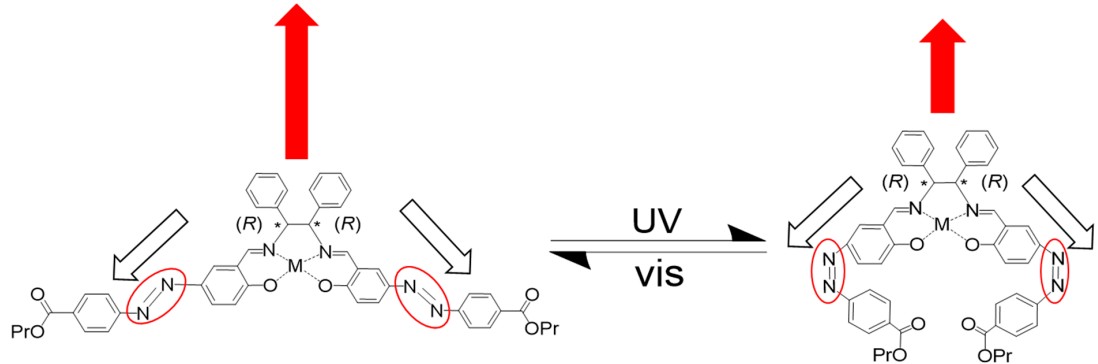

**Figure 11.** Expected shape and dipole changes of *trans*- to *cis*-photoisomerization complexes.

## 4. Discussion

### 4.1. Proposed Contacting Features between AuNP and Chiral Complexes

To begin a full interpretation of the possible behavior of chiral metal complexes and their relationship between AuNP and material-light interactions, experimental facts should be summarized again with the proposed mechanism. The fact that decrease or increase in the CD bands around the plasmon region could be observed may be attributed to the angles of magnetic dipole moment (AuNP)

and electric dipole moment (chiral complexes). Considering weakly, namely, non-fixed connection, and weak induction of CD bands, and the lack of bonding groups for the identical ligands support non-bonded interactions, play a predominant role in inducing CD bands. According to the theoretical treatments, perpendicular and parallel arrangements of normal AuNP vector and dipole moments of complexes ascribe to the decrease or increase in the induced CD bands for NiAz (ZnAz) and CuAz, respectively. Slight difference in the coordination geometries may result in such different alignment of molecules at the initial state. In both cases, the intensity of induced CD bands was not strong, which may support the assumption.

### 4.2. Proposed Changes Due to Trans- to Cis-Photoisomerization

After (linearly polarized or at least non-polarized, namely natural) UV light irradiation, ligand azo-groups undergo photoisomerization to the *cis*-form accompanied by drastic change in the steric factors of chiral complexes. In addition to the intermolecular contacts caused by molecular shapes, this photoisomerization can result in drastic change in UV-vis and CD spectra around $\pi$-$\pi^*$ and n-$\pi^*$ bands. Indeed, three chiral complexes exhibited intense $\pi$-$\pi^*$ and n-$\pi^*$ bands and their changes for typical azo-compounds. However, such spectral changes cannot be regarded as characteristic proof of change of induced CD, because chiral metal complexes can exhibit spectral changes when they are permitted by sufficient space (free volume) or electronic states (typically "lacking" highly electron-withdrawing groups) for photoisomerization. Furthermore, photoisomerization potentially results in changes (decreasing) in the magnitude of dipole moment for chiral complexes. For direct contact and electrostatic dipole-dipole interaction mechanism, CD spectral changes can be proposed after photoisomerization, although only the latter influence (change of dipole moment) can be in agreement with the situation. Thus, the *cis*-form must be the predominant species on AuNPs after UV light irradiation.

### 4.3. Weigert Effect Caused by Linearly Polarized UV Light

Contrary to the previous study with natural UV light [13], we employed linearly polarized UV light in this study. Similar to organic-inorganic composite materials containing a metal complex and azobenzene methyl methacrylate polymer (PMMA) cast films [36–40], anisotropic alignment of azo-compounds and coexisting molecules (Weigert effect [41,42]) can also be induced by linearly polarized UV light. By alternate irradiation of UV and visible light, selected photoisomerization resulted in anisotropic alignment of molecular long axis perpendicular to the electric vector of polarized light (Figure 12, left) [43–46]. Beyond the original Weigert effect, helical alignment of azo-compounds to induce CD bands of supramolecular chirality can be induced by circularly polarized UV light [47]. Hence, when chiral metal complexes (*cis*-forms) can be rotated on the surface of AuNP without changing angles between angles of AuNP (normal vector) and complexes (dipole moment) (Figure 12, right).

It should be mentioned that (even if they are achiral) some dyes in crystalline materials can emerge or cancel induced CD bands when these dye molecules align properly on the surface of the materials [48,49]. The fact suggests that in-plane rotational displacement of mutual arrangement for complexes can result in the appearance of induced CD bands accompanying the Weigert effect. However, the surface of AuNP is not flat but spherical, and the transition moments of complexes are impossible to align appropriately to appear such as induced CD signals.

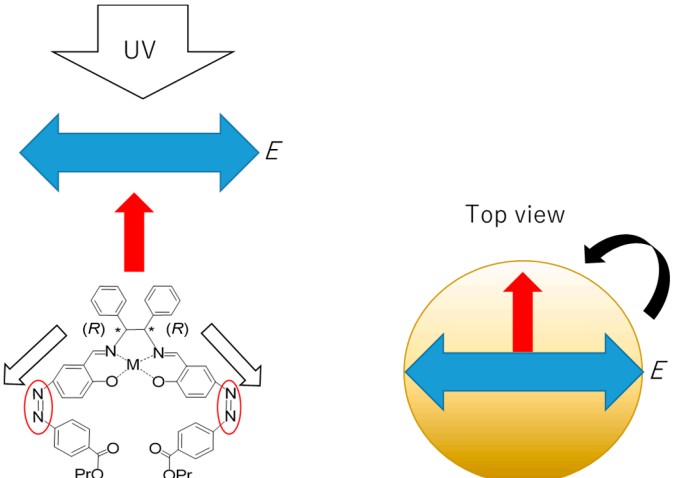

**Figure 12.** Spatial relationship of the Weigert effect. (**Left**) Polarized light and chiral complex in space. (**Right**) Polarized light and chiral complex on the surface.

### 4.4. Dipole-Dipole Interaction within the Exciton Framework

In the exciton framework [50], considering supramolecular chirality [51,52], dipole-dipole interactions of azo-compounds may be important for metal complexes' chirality [53,54]. In the case of hybrid materials in polymer films, in which two components (and accompanying dipole vectors for these components) can freely rotate along the circumstance direction in the plane perpendicular to the light propagation vector, circularly polarized light could induce supramolecular chirality of the analogous metal complexes with azo-compounds [55,56]. In this case, the initial state of weak adsorption resulted in limited chiral metal complex motion. One vector (complex) can rotate in the plane AuNP's surface, while another vector (AuNP) is fixed, because of its permanent nature, to approximately keep their orthogonality. However, "orthogonality" was still kept, treatment within the exciton framework can reasonably describe the spatial relationship and their spectral changes before and after linear UV light irradiation (Weigert effect). Therefore, as schematically depicted in Figure 13, the spatial relationship of chiral metal complexes (and its dipole moment) against (normal vector of) the surface of AuNP should be summarized according to experimental facts and an adaptable theoretical framework.

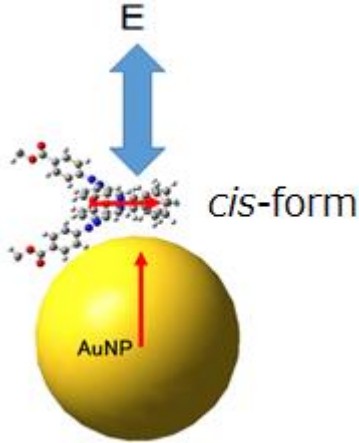

**Figure 13.** Proposed spatial relationship to be treated by the exciton framework or electric vector of the polarized light and chiral complex and the surface of AuNP.

## 5. Conclusions

In summary, we prepared new chiral Schiff base NiAz, CuAz, and ZnAz complexes having azo-groups and *n*-propyl terminals. Weakly (positive and negative) induced CD bands in the plasmon region of AuNP suggest out-of-parallel arrangement of two dipole moments. After linearly polarized UV light irradiation, changes in induced CD spectra can be elucidated by separating the contribution of *cis-trans* photoisomerization and Weigert effect of azobenzene moiety. The resulting CD spectral changes and the initial adsorption features allowed us to estimate the two vectors of dipoles and chiroptical behavior, which are in agreement with the treatment of the exciton framework.

In-plane rotation of chiral complexes to align on a surface [57,58] is deduced in this study. However, on the surface of gold nanoparticles, proximity field light by helical light has been previously reported [59–61]. As in helical axisymmetric in the crystal, the effect of light with chirality in the third axial direction (along the propagation vector of light) will be induced to azobenzene [62–64]. If it can be combined with the surface of the metal nanomaterial, it would be possible to discuss various free operations.

**Author Contributions:** Conceptualization, T.A.; investigation, N.S.; writing—original draft preparation, T.A.; reviewing, T.H.

**Funding:** This research received no external funding.

**Acknowledgments:** The authors thank Yuki Tsutsumi and Ikuma Jahana (students in our groups) for their assistance in performing some experiments.

**Conflicts of Interest:** The authors declare no conflict of interest.

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
