# Peer review of "Orientation of Chiral Schiff Base Metal Complexes Involving Azo-Groups for Induced CD on Gold Nanoparticles by Polarized UV Light Irradiation"

_symmetry, doi:10.3390/sym11091094_

Round 1

Reviewer 1 Report

see attached

Author Response

August 20, 2019

Dear The editor of Symmetry:

Please find an attached revised manuscript of article symmetry-573957 entitled “Orientation of chiral Schiff base metal complexes involving azo-groups for induced CD on gold nanoparticles by polarized UV light irradiation” (correction was marked red letters) and consider answers for query below.

1 Because unidentified solvent molecules were also contained in the powder samples, experimental values were not in agreement with theoretical values assumed as these identical structures (without solvent molecules). This is also associated with the next query, and it is mentioned in the text.

2 Adsorption structure were proposed according to CD and previous studies [13, 14]. Moreover, coordination geometries, namely coordination of solvent molecules, is not clear, though possibility of axial coordination in the solution may be high according to conventionally known "solution paramagnetism" of Schiff base (Ni(II)) complexes. So it was mentioned in the text. However, presence of axial ligands may not influence on induced CD bands.

3 According to previous studies [13, 14], the difference of intensity of d-d (weak) and (induced) Plasmon bands are different each other.

4 Similar to the answer for the previous query (3), interligand transitions (pi-pi and n-pi) associated with azobenzene is quite strong bands, so change of induced CD bands may be weak and different wavelength. In addition, if photoisomerization can be happened smoothly, induced CD bands strongly depends on adsorption structures of chiral complexes. Which are in agreement with the previous studies [13, 14] and makes difficult to exhibit direct proof in this case.

5 Line 25-28 is general explanation of CD, so these sentences were removed.

6 Line 177-181 were fixed as follows:

“As depicted in Figure 11 schematically, trans to cis photoisomerization of salen-type complexes resulted in not only difference of steric features of chiral complexes and their adsorption forms but also the difference of electrostatic properties (decreasing of dipole moment). In contrast, trans-type Schiff base complexes may keep them or at least exhibit small differences because of their symmetry.”  

Major: Only ZnAz complex is diamagnetic, does not show d-d transition, and may afford a slightly distorted (compressed) tetrahedral coordination geometry. Therefore, agreement of calculated spectra, spectral feature of visible region, and change of induced spectra (caused by adsorption form) may be different from other ones.

Minor:

Thank you. Figure 8 was fixed.

According to previous study [13, 14], at least photoismerization completed within 3 min generally, which were also confirmed several times.

That's all.  

English language editing will be requested to MDPI after acceptance (confirming all the text) of this article, if needed.

We would like to say thanks in advance for your prompt treatment of our manuscript.

Best regards,

Prof. Dr. Takashiro Akitsu

Department of Chemistry, Faculty of Science, Tokyo University of Science

1-3 Kagurazaka, Shinjuku-ku, Tokyo 162-8601, Japan

Tel. +81-3-5228-8271, Fax. +81-3-5261-4631

Reviewer 2 Report

Comments to the author:

In this manuscript, the authors presented the synthesis and characterization of new chiral salen type Schiff base (Ni(II), Cu(II), and Zn(II))complexes containing azo-groups absorbed on 10 nm gold nanoparticles (AuNPs). The authors studied the chiral response (theoretically and experimentally) of the complexes individually and in the presence of AuNPs and they suggested out-of-parallel arrangement of two dipole moments. Moreover, they present the variation in circular dichroism (CD) after linearly polarized UV light irradiation of the complexes and they elucidate the mechanism by separation of the contribution of cis-trans photoisomerization and Weigert effect of azobenzene moiety. Overall, I would recommend the publication of this contribution because it presents a novel and facile study which could be used to develop a wide range of novel materials.

The following are some questions and suggestions for improving their work:

Major issues:

When comparing the experimental and the simulated CD and UV-vis spectra of ZnAz there is clear differences such as in the experimental CD spectra the chiral response is almost zero and the simulated spectra is very high. Furthermore, the simulated UV-vis spectra predicts a broad band centered at 350nm, whereas in the experimental spectra the band is centered around 420nm. Could the authors explain this behavior? What happens for the other two complexes? When comparing the UV-vis spectra before and after UV irradiation there is no clear differences neither the complexes nor the complexes in the presence of gold nanoparticles. Could the authors explain this behavior? The CD response between the three complexes is considerably different. Being the CD response of ZnAz extremely weak as compare to the other two complexes. Could the authors explain this behavior?

Minor issues:

Simulated CD and UV-vis spectra should be also presented for NiAz and CuAz complexes. In figure 8, as opposite to the rest of the figures, UV-vis spectra is shown before the CD spectra. After how much time of irradiation were the CD spectra collected? Is there any variation with time?

Author Response

(The authors gave the same response as above.)

Round 2

Reviewer 1 Report

I agree with the changes the authors have made and recommend publication of this manuscript in Symmetry.

Author Response

August 23, 2019

Dear The editor of Symmetry:

Please find an attached the second revised manuscript of article symmetry-573957 entitled “Orientation of chiral Schiff base metal complexes involving azo-groups for induced CD on gold nanoparticles by polarized UV light irradiation” (correction was marked violet letters) and consider answers for query below.

I agree with the changes the authors have made and recommend publication of this manuscript in Symmetry.

@ Thank you.

When comparing the experimental and the simulated CD and UV-vis spectra of ZnAz there is clear differences such as in the experimental CD spectra the chiral response is almost zero and the simulated spectra is very high. Furthermore, the simulated UV-vis spectra predicts a broad band centered at 350nm, whereas in the experimental spectra the band is centered around 420nm. Could the authors explain this behavior? What happens for the other two complexes?

@ Because of difficulty of treatment of excited states by any methods of theoretical calculations such as DFT and semi-empirical methods of normal accuracy, simulated CD and UV-vis spectra may be reliable in a qualitative sense. Therefore, there is a limit to quantitative discussion as well as detailed comparison between experimental and simulated CD and UV-vis spectra.

If due to unidentified solvent molecules experimental values are not in agreement with the theoretical ones is it not clear for me the purpose of the study. What about the other complexes?

@ Generally, spectroscopic study in solutions is treat compounds as they are in solutions, which is not necessarily identical to forms as isolated in crystalline solid-state. The reasons may be crystalline solvents (or as axial ligands in a crystal) can be easily released from as dissolved into solutions. Moreover, molecular conformation of a certain metal complex like the present ones having long pendant ligands in a crystal can flexibly change into other conformation in a solution. From a viewpoint of heat of formation and intermolecular interactions, molecular conformation in solution is more stable than that in a crystal. Experimentally, however, spectra in solutions and molecular structures distorted by crystal packing can be obtained only, therefore computational simulation plays an important role in discussing the (assumed) molecular forms in solutions to compare with or discuss spectra in solutions.

   Additionally, in order to the numbers of identified solvent molecules for each complex, elemental analysis was re-calculated as the corresponding hydrates to obtain reasonable results.

When comparing the UV-vis spectra before and after UV irradiation there is no clear differences neither the complexes nor the complexes in the presence of gold nanoparticles. Could the authors explain this behavior?

@ As suspension-like situation rather than clear solutions, UV-vis spectra less sensitive than CD spectra (intense induced plasmon region) to detect conformational changes of chiral metal complexes.

Simulated CD and UV-vis spectra should be also presented for NiAz and CuAz complexes.

@ According to your comment, these spectra were added in Figure 2.

That's all.

English language editing will be requested to MDPI after acceptance (confirming all the text) of this article, if needed.

We would like to say thanks in advance for your prompt treatment of our manuscript.

Best regards,

Prof. Dr. Takashiro Akitsu

Department of Chemistry, Faculty of Science, Tokyo University of Science

1-3 Kagurazaka, Shinjuku-ku, Tokyo 162-8601, Japan

Tel. +81-3-5228-8271, Fax. +81-3-5261-4631

Reviewer 2 Report

The authors has not answered all the major/minor issues requested. The following are still not answered:

Major issues:

When comparing the experimental and the simulated CD and UV-vis spectra of ZnAz there is clear differences such as in the experimental CD spectra the chiral response is almost zero and the simulated spectra is very high. Furthermore, the simulated UV-vis spectra predicts a broad band centered at 350nm, whereas in the experimental spectra the band is centered around 420nm. Could the authors explain this behavior? What happens for the other two complexes? If due to unidentified solvent molecules experimental values are not in agreement with the theoretical ones is it not clear for me the purpose of the study. What about the other complexes? When comparing the UV-vis spectra before and after UV irradiation there is no clear differences neither the complexes nor the complexes in the presence of gold nanoparticles. Could the authors explain this behavior?

Minor issues:

Simulated CD and UV-vis spectra should be also presented for NiAz and CuAz complexes.

Author Response

(The authors gave the same response as above.)

Round 3

Reviewer 2 Report

The authors have satisfactorily answered all the questions raised therefore I recommend the publication of this manuscript in Symmetry.